# PanoWan: Lifting Diffusion Video Generation Models to 360° with Latitude/Longitude-aware Mechanisms

**Yifei Xia**[1,2,3]* **Shuchen Weng**[4]* **Siqi Yang**[5] **Jingqi Liu**[1,2] **Chengxuan Zhu**[6]
**Minggui Teng**[1,2] **Zijian Jia**[7] **Han Jiang**[3] **Boxin Shi**[1,2]†

[1]State Key Lab of Multimedia Info. Processing, School of Computer Science, Peking University
[2]Nat'l Eng. Research Ctr. of Visual Tech., School of Computer Science, Peking University
[3]OpenBayes Information Technology Co., Ltd. [4]Beijing Academy of Artificial Intelligence
[5]Institute for Artificial Intelligence, Peking University
[6]Nat'l Key Lab of General AI, School of Intelligence Science and Technology, Peking University
[7]School of Artificial Intelligence, Beijing University of Posts and Telecommunications
`{yfxia,shuchenweng,yousiki,peterzhu,minggui_teng,shiboxin}@pku.edu.cn`
`liujingqi@stu.pku.edu.cn   jiazijian@bupt.edu.cn   hahn@openbayes.com`

## Abstract

Panoramic video generation enables immersive 360° content creation, valuable in applications that demand scene-consistent world exploration. However, existing panoramic video generation models struggle to leverage pre-trained generative priors from conventional text-to-video models for high-quality and diverse panoramic videos generation, due to limited dataset scale and the gap in spatial feature representations. In this paper, we introduce PanoWan to effectively lift pre-trained text-to-video models to the panoramic domain, equipped with minimal modules. PanoWan employs latitude-aware sampling to avoid latitudinal distortion, while its rotated semantic denoising and padded pixel-wise decoding ensure seamless transitions at longitude boundaries. To provide sufficient panoramic videos for learning these lifted representations, we contribute PANOVID, a high-quality panoramic video dataset with captions and diverse scenarios. Consequently, PanoWan achieves state-of-the-art performance in panoramic video generation and demonstrates robustness for zero-shot downstream tasks. Our project page is available at `https://panowan.variantconst.com`.

## 1 Introduction

Text-based panoramic video generation aims to produce a complete 360° view, ensuring coherent spatial and visual relationships between elements within the scene. Such inherent property is highly valuable for conventional VR content, the construction of interactive game worlds [35, 8], and the simulation of environments for embodied AI [18].

The remarkable capabilities of conventional text-to-video models [28, 33, 4] motivate researchers to leverage their generative priors to panoramic video generation. One intuitive strategy generates local perspective conventional videos and integrates them during inference. While these training-free methods [12, 21] entirely preserve generative priors, they sacrifice overall consistency as they struggle to establish cross-view long-range dependencies. Alternatively, fine-tuning conventional text-to-video models [32, 41] also faces challenges. On one hand, existing panoramic video datasets are limited in scope and scale compared to conventional ones. On the other hand, the gap in spatial

---

*Equal contribution.
†Corresponding author.

39th Conference on Neural Information Processing Systems (NeurIPS 2025).

**Text-to-video generation**

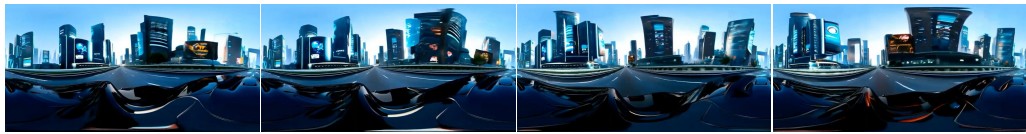

*A black hyper-car speeds through cyberpunk highway.*

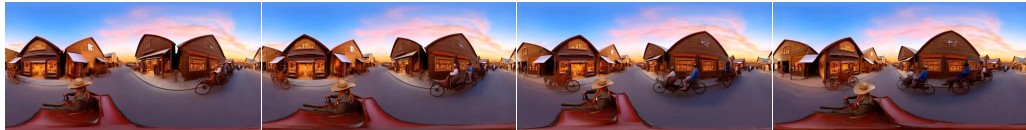

*Cowboys ride through sunset-lit western town, visitors explore old streets.*

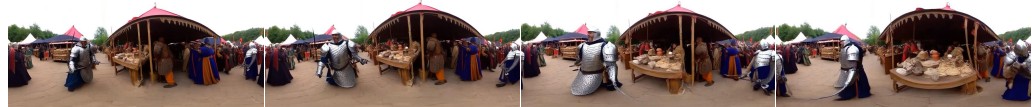

*Medieval event with knights jousting, crowds cheering, camps bustling lively.*

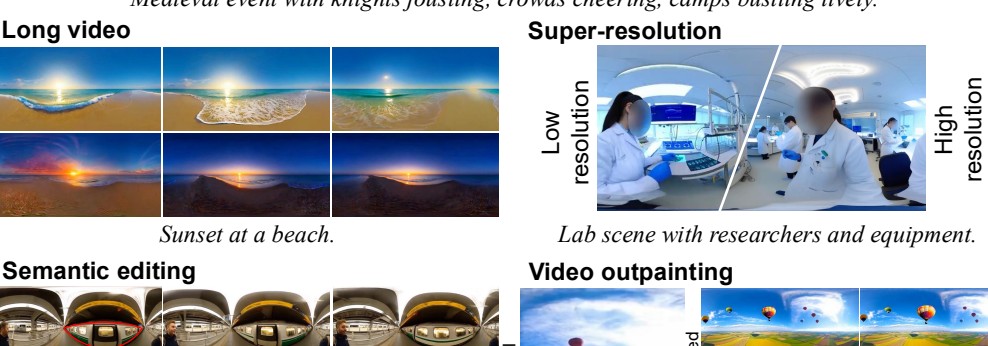

**Long video**

*Sunset at a beach.*

**Super-resolution**

*Lab scene with researchers and equipment.*

**Semantic editing**

*Change the color of the train to red.*

**Video outpainting**

*Colorful hot air balloons.*

Figure 1: PanoWan is a text-based panoramic video generation framework. It lifts pre-trained generative priors from a conventional text-to-video model to the panorama, and enables generating diverse scenarios for long videos. Equipped with training-free techniques, PanoWan supports zero-shot editing of panoramic videos, including super-resolution, semantic editing, and video outpainting.

representations (*e.g.*, latitudinal distortions and seam longitudes) between panoramic and conventional videos potentially hinder the effective prior leverage from pre-trained conventional models.

In this paper, we pursue a training-based approach to overcome current bottlenecks. We propose **PanoWan**, a framework for **Pano**ramic video generation based on **Wan** 2.1 [33]. Equipped with minimal modules, PanoWan effectively lifts generative priors from a pre-trained conventional text-to-video model to the panorama. To bridge the gap in spatial feature representations between panoramic and conventional videos, we design latitude-aware sampling to address latitudinal distortions caused by equirectangular projection. Since text-to-video models lack continuity awareness for left and right boundaries, we achieve seamless longitude transitions using the rotated semantic denoising to address semantic inconsistency and the padded pixel-wise decoding to resolve pixel-wise disharmony.

As learning to lift spatial representations from conventional videos to the panorama requires large-scale data, we further introduce PANOVID. This **Pano**ramic **Vid**eo dataset offers diverse scenarios (*e.g.*, landscape, streetscape, and humanscape), includes over 13K captioned video clips totaling 944 hours, and features data processing tailored for panoramic video generation. As shown in Fig. 1, our framework produces panoramic videos from text descriptions, enabling long video generation. Additionally, it enables training-free editing of user-provided panoramic videos, including super-resolution, semantic inpainting, and video outpainting. Extensive experiments demonstrate that

PanoWan achieves state-of-the-art panoramic video generation performance across seven metrics, alongside robust zero-shot capabilities for various downstream tasks.

Our contributions can be summarized as follows:

- We contribute PANOVID, a large-scale and high-quality panoramic video dataset with captioned video clips, tailored for text-based panoramic video generation.
- We propose PanoWan, lifting generative priors from a pre-trained model to show state-of-the-art performance on panoramic video generation and robustness for downstream tasks.
- We integrate the latitude-aware sampling, rotated semantic denoising, and padded pixel-wise decoding to bridge the spatial difference between panoramic and conventional videos.

## 2 Related Works

### 2.1 Video Diffusion Models

Diffusion models have demonstrated impressive results in image generation, but extending them to videos introduces additional challenges (*e.g.*, temporal consistency and computational efficiency). Early models adapt image diffusion models with cascaded architecture [10], temporal layers [25], and attention mechanisms [3]. To further improve efficiency, LVDM [9] introduces a lightweight latent video diffusion model with a 3D latent space and hierarchical structure for long video generation. Meanwhile, SVD [4] improves video diffusion using large and high-quality datasets. Recent Diffusion Transformers (DiTs) [22] effectively model complex spatio-temporal dynamics for video generation, operating on latent space compressed by 3D VAE [39]. After that, more large-scale models (*e.g.*, HunyuanVideo [14], CogVideoX [38], and Seaweed [24]) emerge and demonstrate the benefits of scaling both model and data size. This motivates us to adopt Wan 2.1 [33] as the backbone to leverage its strong generative priors and temporal modeling capabilities for panoramic video generation.

### 2.2 Panoramic Video Generation

**Text conditioned generation.** 360DVD [32] is the pioneer to introduce stable video generation techniques to panoramic video generation by proposing a 360-Adapter and a set of 360 enhancement techniques. Training-free methods (*e.g.*, DynamicScaler [12] and SphereDiff [21]) create panoramic videos by generating local patches and then composing them together into a complete panorama, which inherently break global consistency. With the development of video diffusion models, VideoPanda [36] augments diffusion models with multi-view attention. PanoDiT [41] uses a DiT backbone with global-temporal attention and panoramic-specific losses for coherent long-range generation. Despite these advancements, existing methods still suffer from observable latitude distortions and issues with seam-free longitude transitions. In this work, we introduce PanoWan, a framework that addresses these challenges by lifting generative priors from pre-trained text-to-video models to panorama.

**Image or video conditioned generation.** Imagine360 [26] adopts antipodal-aware motion modeling to convert perspective videos to panoramic views. Building on static panoramas, 4K4DGen [16] lifts them into dynamic 4D scenes via spatial-temporal denoising. HoloTime [42] further leverages Gaussian splatting and a two-stage diffusion process for high-fidelity 4D reconstruction. VidPanos [20] treats panorama generation as a space-time outpainting task from panning video inputs, while Argus [19] integrates motion and geometry cues for enhanced video-to-360° synthesis. These explorations highlight a trend toward unifying spatial, temporal, and geometric reasoning. We demonstrate that PanoWan possesses these capabilities, with robust zero-shot capabilities for downstream tasks.

## 3 PANOVID Dataset

The absence of paired datasets has long been regarded as one of the primary barriers to advancing the performance of panoramic video generation models [32]. Existing text to panoramic video generation methods [32, 36, 41] mainly rely on WEB360 dataset [32], which contains only 2114 video clips of 10 seconds each. Although Argus [19] filters out over 283K video clips from the 360-1M dataset [29], it is not built for the text-based panoramic video generation task, providing no paired captions, and showing significant distribution bias for the scenario semantics.

To address these limitations, we present PANOVID, a large-scale and high-quality dataset with diverse scenarios and balanced semantics, tailored for text-based panoramic video generation. Our data collection process begins by aggregating videos from existing panoramic sources, including 360-1M [29], 360+x [5], Imagine360 [26], WEB360 [32], Panonut360 [37], the Miraikan 360-degree Video Dataset [1], and a public dataset of immersive VR videos [15]. These sources cover both large-scale web collections (with rich but noisy YouTube content) and more curated institutional datasets (with higher quality but limited size).

**Vision-language-based filtering pipeline.** To transform these heterogeneous sources into a high-quality paired dataset, we design a scalable five-stage filtering pipeline guided by a vision-language model. *(i) Initial filtering by popularity:* We first filter large-scale collections such as 360-1M [29], retaining only videos with at least 1000 views to discard low-quality or trivial content. *(ii) Shot segmentation:* Each raw video is segmented into 10-second continuous clips using PySceneDetect, ensuring scene consistency and temporal coherence. *(iii) Vision-language annotation:* We employ Qwen-2.5-VL [2] to process each clip, generating a descriptive caption and predicting the associated POI (Point-of-Interest) category in a structured JSON format. Notably, we retain only clips that the model identifies as true ERP (equirectangular projection) panoramic videos. *(iv) Motion score filtering:* Following [7], we compute a normalized optical-flow magnitude and remove clips with a motion score below 0.4 to exclude static or nearly still scenes. *(v) Aesthetic score filtering:* Finally, we utilize Q-Align [34] to evaluate the aesthetic quality of each frame, discarding clips with a mean aesthetic score below 3.0. This ensures that the dataset maintains a consistent level of visual quality. This hierarchical filtering process effectively removes noise while preserving diversity and realism, yielding panoramic video clips that are semantically meaningful, visually appealing, and motion-rich. It also enables large-scale automatic caption generation without manual annotation cost.

**Semantic balancing.** We observe that the automatically annotated POI categories exhibit strong long-tail distribution: natural scenes such as *Mountains* and *Parks* dominate, while indoor and human-centric environments (*e.g.*, *Libraries*, *Shops*, *Theaters*) are underrepresented. To alleviate this imbalance and enhance the generalization of models trained on PANOVID, we cap the maximum number of clips per POI category to 200, selecting the highest-ranking clips based on a combination of aesthetic and motion scores. All clips from underrepresented categories are fully retained. This strategy prevents overfitting to dominant scene types and ensures a diverse representation across environments.

**Dataset statistics and characteristics.** After the filtering, balancing, and a final deduplication step based on caption similarity, PANOVID comprises over 13K high-quality panoramic video clips, totaling approximately 944 hours of content. Compared with prior datasets, PANOVID offers not only substantially larger scale but also structured text–video pairs with fine-grained POI categories and balanced semantic coverage. It thereby provides a robust foundation for training and evaluating text-conditioned panoramic video generation models.

# 4 Method

Panoramic videos have a different spatial feature representation compared to conventional ones. Inspired by GEN3C [23], we effectively preserve the generative prior of pre-trained models by equipping minimal modules and fine-tuning a small subset of parameters via LoRA [11]. We firstly introduce our video diffusion backbone and formulate the spherical coordinate mapping (Sec. 4.1). Next, we propose the latitude-aware sampling to avoid latitude distortion, along with its corresponding analysis (Sec. 4.2). Finally, we present the rotated semantic denoising and the padded pixel-wise decoding to achieve the seamless longitude transitions (Sec. 4.3).

## 4.1 Preliminaries

**Video diffusion models.** We employ Wan 2.1 [33] as the video generation backbone, with spatial-temporal Variational AutoEncoders (VAEs) to map high-dimensional videos into compact latent codes. The flow matching framework [17] is used to model a unified denoising diffusion process. Specifically, given a clear video $x$, a VAE encoder $\mathrm{E}(\cdot)$ first projects the video into the latent space $z_1 = \mathrm{E}(x)$. During training, a noise $z_0 \sim \mathcal{N}(0, I)$ is sampled, and an intermediate latent code $z_t = tz_1 + (1 - t)z_0$ is constructed by linearly interpolating between $z_1$ and $z_0$ at timestep $t \in [0, 1]$.

The training goal is to predict the ground truth velocity $v_t = \mathrm{d}z_t/\mathrm{d}t = z_1 - z_0$, and the loss function is formulated as:

$$\mathcal{L} = \mathbb{E}_{z_0, z_1, c_{\text{txt}}, t} ||u(z_t, c_{\text{txt}}, t; \theta) - v_t||^2, \tag{1}$$

where $c_{\text{txt}}$ is the text embedding, $\theta$ is the parameters of the prediction model, and $u(z_t, c_{\text{txt}}, t; \theta)$ is the predicted velocity of the model.

**Spherical coordinate mapping.** Panorama captures a 360° view, inherently representing signals in spherical coordinates $(\varphi, \theta)$. To leverage generative priors from conventional images and videos that operate in Cartesian coordinates $(x, y)$, we employ the equirectangular projection (ERP) $\mathcal{P}_{\text{ERP}}$ to map between these coordinate systems for panoramic videos:

$$\mathcal{P}_{\text{ERP}} : [2R] \times [R] \to [0, 2\pi] \times [-\frac{\pi}{2}, \frac{\pi}{2}], \quad (x, y) \mapsto (\varphi, \theta) = \left( \frac{2x+1}{2R}\pi, \frac{2y+1-R}{2R}\pi \right), \tag{2}$$

where $R$ is the radius of the sphere. $\varphi$ and $\theta$ are longitude and latitude respectively. While ERP enables the direct application of pre-trained VAEs to encode panoramic videos into latent codes for diffusion processes, it introduces extreme horizontal stretching in polar regions. This horizontal stretching phenomenon arises from the altered representation of distances during projection, and is recognized by changes in horizontal signal frequency. Let $\mathrm{d}s_\varphi$ and $\mathrm{d}s_\theta$ represent the infinitesimal arc lengths along lines of constant latitude and longitude, respectively. They are formulated as:

$$\mathrm{d}s_\varphi = 2R \arcsin(\cos\theta \cdot \sin\frac{\mathrm{d}\varphi}{2}) = R\cos\theta\,\mathrm{d}\varphi, \quad \mathrm{d}s_\theta = R\,\mathrm{d}\theta, \tag{3}$$

where $\theta$ is the latitude. We further consider the spherical frequency $f_{\text{sph}}$ (cycles per unit physical distance) and the Cartesian frequency $f_{\text{car}}$ (cycles per pixel in the image). Assuming that warping preserves content, their relationship in frequency is scaled by the change in distance:

$$f_{\text{car},y}(x) = f_{\text{sph},\theta}(\varphi)\frac{\mathrm{d}s_\theta}{\mathrm{d}\theta} = R f_{\text{sph},\theta}(\varphi), \quad f_{\text{car},x}(y) = f_{\text{sph},\varphi}(\theta)\frac{\mathrm{d}s_\varphi}{\mathrm{d}\varphi} = R f_{\text{sph},\varphi}(\theta)\cos\theta. \tag{4}$$

Consequently, in polar regions ($|\theta| \approx \frac{\pi}{2}$, namely $y \approx 0$ or $y \approx R$), the horizontal frequency in the Cartesian coordinate becomes near-zero ($f_{\text{car},x}(y) \approx 0$). Such distortion in the horizontal frequency distribution significantly degrades the effectiveness of transferring priors.

## 4.2 Latitude-Aware Mechanisms

**Latitude-aware sampling.** Conventional text-to-video models typically assume independent and identically distributed (i.i.d.) Gaussian noise vectors for each Cartesian coordinate $(x, y)$. To avoid latitudinal distortion in polar regions of ERP, we propose the latitude-aware sampling to better align the initial noise with the spherical frequency distribution for panoramic video generation. As illustrated in the top-left of Fig. 2, our latitude-aware sampling remaps the horizontal sampling coordinates based on latitude to preserve frequency consistency across the sphere. Specifically, after initializing the latent map with i.i.d. Gaussian noise vectors, we calculate the sampling noise by remapping the horizontal sampling coordinate $x$ based on the latitude corresponding to row $y$, and then applying the interpolation:

$$P'(x, y) = \text{Interp}_P\left( R + (x - R)\cos(\frac{2y+1-R}{2R}\pi), y \right), \tag{5}$$

where $P'(x, y)$ is the interpolated noise vector at coordinate $(x, y)$. $\text{Interp}(\cdot)$ is formulated as the interpolation function for normalization:

$$\text{Interp}_P(x, y) = \text{sgn}(\text{BI}(P, x, y))\sqrt{\text{BI}(P^2, x, y)}, \tag{6}$$

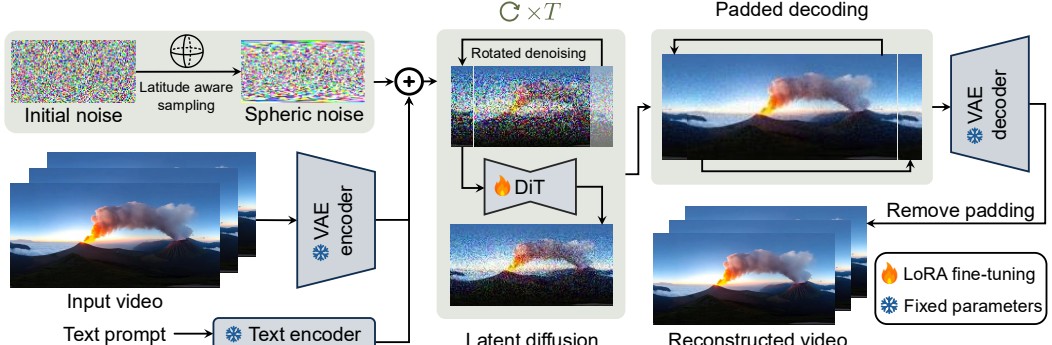

Figure 2: The pipeline of our proposed PanoWan, aware of spherical coordinates. To avoid latitudinal distortion, initial random Gaussian noise is remapped to align with the spherical frequency distribution using the latitude-aware sampling (Sec. 4.2). Next, this remapped noise serves as the latent code within the VAE-encoded latent space. A DiT-based denoising network then iteratively refines this latent representation, where rotated denoising is applied by rolling the latent grid to ensure semantic consistency across longitudinal boundaries. After that, padded pixel-wise decoding provides the VAE decoder with extended context, enabling the mapping of the denoised latent code back into seamless panoramic videos (Sec. 4.3). The DiT backbone within PanoWan is efficiently fine-tuned using LoRA, where most parameters of the pre-trained text-to-video model remain frozen to preserve its strong generative priors.

where $\mathrm{sgn}(\cdot)$ is the sign function, and $\mathrm{BI}(P, x, y)$ is the standard bilinear interpolation for vector $P$ at coordinate $(x, y)$. Consequently, the resulting noise vectors sampled by our strategy preserve $\mathbb{E}[P'(x, y)] = 0$ and $\mathbb{E}[\mathrm{Var}\, P'(x, y)] = 1$, approaching the distribution on which the diffusion models are pre-trained. The proof is given in the supplementary materials.

**Frequency domain analysis.** We aim to prove that the horizontal frequency properties of the proposed sampling correctly represent the inherent properties of the spherical coordinate, following the methodology of 1-D Discrete Fourier Transform (DFT). Denote the maximal frequency of the original signal in the spherical space as $f_{\max}$, and the maximal frequency in the Cartesian coordinates at the latitude of $\theta$ is determined by:

$$\max f_{\mathrm{car}, x}(y) = \max R \cos \theta \cdot f_{\mathrm{sph}, \varphi}(\theta) \leq R f_{\max}, \tag{7}$$

where the equation only holds when $\theta = 0$, namely on the equator. For the proposed design, it guarantees $\max f_{\mathrm{car}, x} = 2R$ along the equator as $\cos(\theta) = 1$ and the original pixels along $P(\cdot, \frac{R-1}{2})$ is taken. Therefore, the maximal spherical frequency is $f_{\max} = 2$. According to Eq. (5), the warped results only depends on the values between $(R - R \cos(\theta), y)$ and $(R + (R-1) \cos(\theta), y)$ in the original Cartesian grid $P$. According to DFT, the support of the spectrum is reduced to:

$$\lceil R \cos(\theta) \rceil + \lceil (R-1) \cos(\theta) \rceil \approx 2R \cos \theta = f_{\max} R(\theta) \cos \theta = \max f_{\mathrm{car}, x}(y), \forall \theta \in [-\frac{\pi}{2}, \frac{\pi}{2}]. \tag{8}$$

This matches the inherent property of the panorama in the frequency domain in every latitude.

## 4.3 Longitude Continuity Mechanisms

**Seamless longitude transitions.** Pre-trained conventional text-to-video models lack continuity awareness required between the columns of left and right boundaries. Consequently, applying their generative priors directly for panoramic video generation leads to seam artifacts, resulting in an observable transition where the easternmost and westernmost longitudes meet. To achieve seamless longitude transitions, we recognize that these artifacts arise from both semantic inconsistency and pixel-wise disharmony. This motivates us to propose the rotated denoising and the padded decoding to significantly remove the artifacts.

**Rotated semantic denoising.** The video generation backbone inherently introduces semantic inconsistency at each denoising step. Since the pre-trained generative priors lack the continuity awareness, the semantic in leftmost and rightmost longitudes is typically inconsistent, which are further accumulated during the iterative denoising steps and finally produce an obvious transition.

Our proposed rotated semantic denoising aims to spread the transition error evenly to different longitudes. Let $\mathcal{R}_{s_t}(\cdot)$ be the circular-shift operator and $W$ denote the width of the latent code. As shown in Fig. 2, we horizontally roll the latent code $Z_t$ by $\{s_t = t \bmod W\}$ columns at denoising step $t$ and then undo the shift:

$$Z_{t+1} = \mathcal{R}_{-s_t}\Big(\phi_\theta\big(\mathcal{R}_{s_t}(Z_t)\big)\Big),\tag{9}$$

where $\phi_\theta(\cdot)$ is the noise predictor. As a result, the inherent accumulative error for horizontal coordinate $x$ after $T$ denoising steps is:

$$E_T(x) = \sum_{t=1}^{T} \varepsilon_t\big((x + s_t) \bmod W\big),\tag{10}$$

where $\varepsilon_t(\cdot)$ is prediction error for the transition and step $t$, which would concentrate at a fixed seam if no rotation are applied. Due to the rotation strategy, this error at physical coordinate $x$ at step $t$ is determined by the logical position $\{(x + s_t) \bmod W\}$. Over $T$ steps, these logical coordinates $\{(x + s_t) \bmod W\}_{t=1}^{T}$ ideally approach a uniform permutation for all longitudes. This effectively suppresses seam artifacts by a factor approaching $1/W$.

**Padded pixel-wise decoding.** When decoding latent codes back to the pixel space, the pre-trained VAE decoder D often introduces pixel-wise inconsistencies, as it is trained on conventional videos and lacks awareness of the spatial continuity required across the left-right seam of panoramic videos [19]. Inspired by previous works [30, 40], we present the padded pixel-wise decoding. Let $Z_0$ be the denoised latent code. We first create a padded latent code $Z_0' = P_r(Z_0)$, where $P_r(\cdot)$ is a circular padding operator that extends $Z_0$ by $r$ columns of context on the side, and the content at horizontal coordinate $x$ is $\{x \bmod W\}$ in $Z_0$. Finally, we center crop the decoded panoramic videos after the decoding $V = \text{Crop}\big(\text{D}(Z_0')\big)$, as illustrated in Fig. 2. This approach ensures that pixels near the original seam boundaries are decoded with $r$ columns of horizontal panoramic context. Consequently, the VAE decoder can effectively leverage its generative priors learned from conventional videos to avoid the seam artifacts.

## 5 Experiments

### 5.1 Training Details

PanoWan is built on Wan 2.1-1.3B-T2V [33] as the video generation backbone. We train PanoWan at a resolution of $448 \times 896$, closely matching the pre-trained resolution of this backbone model. For parameter-efficient training, LoRA [11] with a rank of 64 is applied to the query, key, value, and output projections of the attention mechanisms, as well as to the feed-forward networks. The model is trained for 200K iterations on our contributed PANOVID dataset. The training process employs the AdamW optimizer [13] with a learning rate of $1 \times 10^{-4}$ and a batch size of 8. Training is conducted on 8 NVIDIA H100 GPUs for approximately 18 hours. During each iteration, clips of 81 consecutive frames are randomly sampled from the videos. Consequently, only 21.9M parameters are adjusted, constituting approximately 1.6% of the base model's total parameters.

### 5.2 Panoramic Video Evaluation Metrics

Existing panoramic video generation methods either directly apply conventional video evaluation metrics [12, 41] or rely on subjective user preferences [32], lacking metrics that comprehensively assess both perceptual quality and spherical consistency critical for panoramic video evaluation. This motivates us to adapt general video quality metrics for panoramic videos and to introduce additional panorama-specific metrics for structural properties of 360° content.

**General metrics.** We apply Frechét Video Distance (FVD) [27] to evaluate overall video quality and VideoCLIP-XL [31] to assess text-video alignment. Following DynamicScaler [12], we also

Table 1: Quantitative comparison results of PanoWan and previous text-based panoramic video generation models. ↑ (↓) means higher (lower) is better. Throughout the paper, best performances are highlighted in **bold**.

| Method | General Metrics | | | Panoramic Metrics | | |
|---|---|---|---|---|---|---|
| | FVD ↓ | VideoCLIP-XL ↑ | Image Quality ↑ | End Continuity ↓ | Motion Pattern ↑ | Scene Richness ↑ |
| 360DVD [32] | 1750.36 | 20.27 | 0.7054 | 0.0323 | 5.8% | 6.6% |
| DynamicScaler [12] | 2146.04 | 21.13 | 0.7188 | 0.0339 | 4.0% | 2.6% |
| Ours (W/o LAS) | 1520.69 | 21.20 | 0.7205 | 0.0278 | 16.2% | 19.4% |
| Ours (W/o RSD) | 1302.48 | 21.76 | 0.7243 | 0.0327 | 15.6% | 18.8% |
| Ours (W/o PPD) | 1294.03 | 21.81 | 0.7239 | 0.0294 | 22.0% | 17.4% |
| **Ours (full)** | **1281.21** | **21.86** | **0.7249** | **0.0270** | **36.4%** | **35.2%** |

calculate specific metrics for image quality. To adapt these general metrics for panoramic videos, we project each video onto a cube map and compute metric scores separately on each of the six faces. The final reported score for a video $v$ is a weighted average:

$$\bar{\mathbf{f}}(v) = \sum_{f \in \mathcal{F}} \alpha_f \cdot \Phi\big(\mathcal{P}_f(v)\big), \tag{11}$$

where $\mathcal{F}$ denotes the set of cube map faces, $\mathcal{P}_f(v)$ is the projection of video $v$ onto face $f \in \mathcal{F}$, $\Phi$ is the metric function, and $\alpha_f$ is the weight assigned to face $f$. Following OmniFID [6], we assign weights $\alpha_{\text{top}} = \alpha_{\text{bottom}} = \frac{1}{3}$ and $\alpha_{\text{side}} = \frac{1}{12}$ for each of the four lateral faces.

**Panoramic metrics.** Following previous works [32, 12], we evaluate motion patterns and scene richness with user preferences. We additionally introduce a quantitative metric for evaluating the end continuity of generated panoramic videos, tailored to capture artifacts across longitude boundaries. Specifically, this metric computes the mean absolute pixel difference across the left and right boundaries, directly capturing discontinuities at the longitude seam.

## 5.3 Comparison with State-of-the-art Methods

We evaluate PanoWan against existing text-based panoramic video generation methods, including 360DVD [32] and DynamicScaler [12], on the PANOVID test split containing 67 non-overlapping clips. Quantitatively, as shown in Tab. 1, PanoWan achieves state-of-the-art performance across both general and panoramic metrics (detailed in Sec. 5.2). Qualitatively, we present visual results to highlight our advantages. For instance, DynamicScaler [12] falls short in complex scenarios (Fig. 3, first sample), and 360DVD [32] exhibits notable distortion in polar regions (Fig. 3, second sample). In contrast, PanoWan effectively maintains global consistency and visual coherence, achieving superior performance in generating high-fidelity panoramic videos.

Following 360DVD [32], we also conducted a human-preference study to complement automatic metrics. 25 participants compared videos generated from 50 random test prompts among PanoWan, 360DVD [32], and DynamicScaler [12]. PanoWan was preferred in 71.36% of cases, versus 16.72% and 11.92% for 360DVD [32] and DynamicScaler [12], respectively. These results confirm that the perceptual improvements observed quantitatively are also recognized subjectively.

## 5.4 Ablation Studies

We conduct ablation studies to validate the effectiveness of proposed modules in PanoWan: latitude-aware sampling (LAS), rotated semantic denoising (RSD), and padded pixel-wise decoding (PPD).

**Quantitative results.** As shown in Tab. 1, removing LAS primarily affects general metrics—FVD increases from 1281.21 to 1520.69, and VideoCLIP-XL drops from 21.86 to 21.20—indicating the model struggles to learn panoramic features in high-latitude regions without frequency-aligned noise initialization. In contrast, removing RSD or PPD mainly degrades panoramic metrics (*e.g.*, end continuity increases from 0.0270 to 0.0327 and 0.0294, respectively), confirming their roles in achieving seamless longitude transitions.

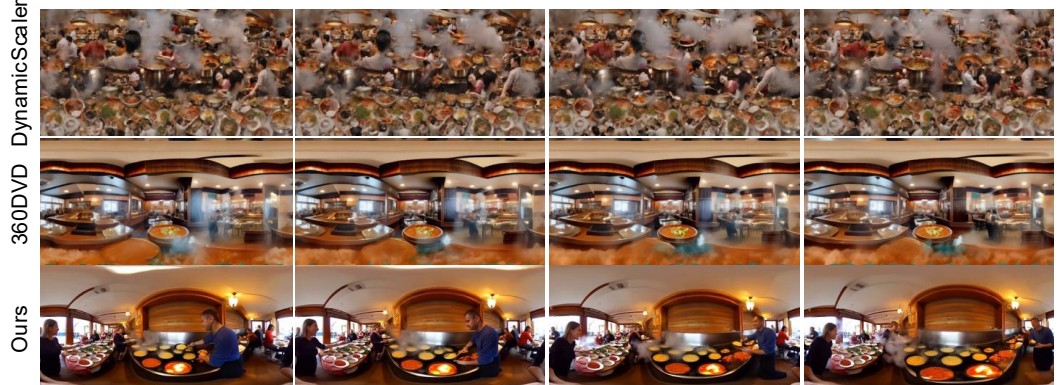

*Wide-angle panoramic interior capturing an energetic hot pot restaurant bustling with diners enthusiastically cooking ingredients in bubbling pots. Colorful plates arranged invitingly, steam rising, spirited conversations filling warm, inviting atmosphere emphasizing community and culinary enjoyment.*

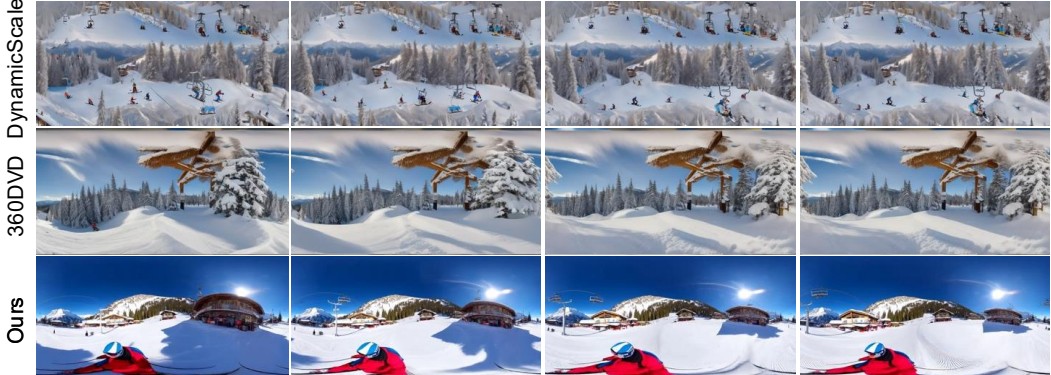

*Expansive panoramic view capturing a vibrant ski resort nestled among towering snowy mountains, as skiers gracefully descend pristine slopes amid cozy alpine chalets. Chairlifts glide leisurely under crisp blue skies while visitors gather around lively outdoor cafes, soaking in the sunny winter scenery.*

Figure 3: Visual comparison results with existing text-based panoramic video generation methods.

**Qualitative results.** We further provide qualitative evaluation in Fig. 4. When generating high-latitude elements like LED panels (which should appear straight in perspective views but are inherently distorted in the equirectangular projection), PanoWan without LAS fails to render them with the correct geometric appearance. When RSD is discarded, semantic inconsistencies become apparent at the longitude seam, due to the lack of mechanism for continuity awareness and the error accumulation during the denoising process. When PPD is removed, observable seam artifacts occur because conventional VAE decoder introduces pixel-wise inconsistencies at boundaries. Consequently, the full PanoWan model with all modules enabled achieves the best performance.

## 5.5 Application

As a text-based panoramic video model, PanoWan shows robust zero-shot capabilities across a wide range of downstream tasks. We present representative examples in Fig. 1 and additional examples in supplementary materials due to the space limitation.

**Long video generation.** To generate panoramic videos longer than the model's native temporal context, we adopt a sliding-window inference strategy in the latent space. At each denoising step, the latent code is partitioned into temporally overlapping chunks, where each chunk is conditioned on a corresponding segment of text prompts expanded by an LLM. Adjacent chunks share overlapping frames that serve as temporal context, and their outputs are fused through a linear blending function on the overlapping regions.

**Super-resolution for panoramic videos.** To generate high-resolution panoramic videos from low-resolution ones, we first encode each low-resolution video into its corresponding latent code. After

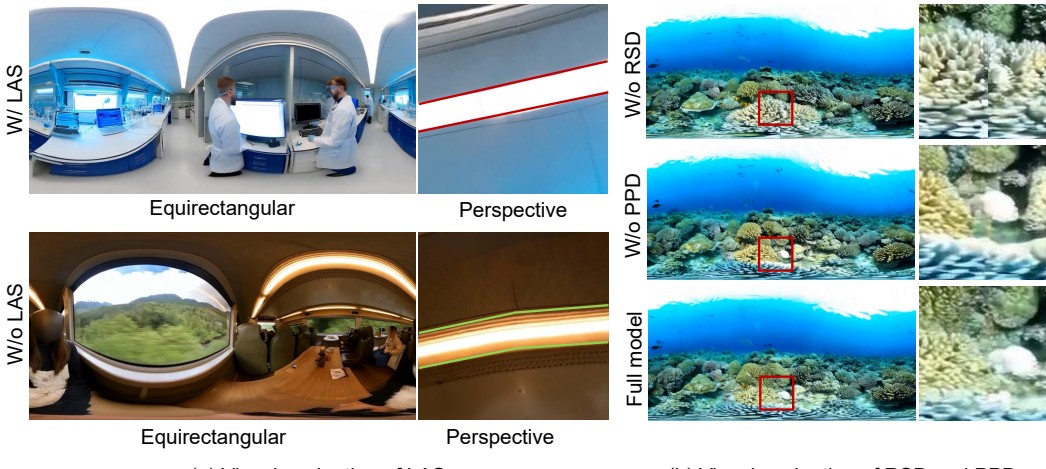

| | |
|---|---|
| (a) Visual evaluation of LAS | (b) Visual evaluation of RSD and PPD |

Figure 4: Qualitative evaluation of proposed latitude/longitude-aware mechanisms. (a) With the proposed Latitude-Aware Sampling (LAS), PanoWan ensures that content generated at high latitudes exhibits an accurate geometry when presented in a perspective view. (b) By combining Rotated Semantic Denoising (RSD) and Padded Pixel-wise Decoding (PPD), PanoWan achieves seamless longitude transitions. For visualization, videos are rolled $180°$ to center the seam.

injected noise, the latent code is denoised based on user-provided text descriptions, producing results with structural consistency and visual fidelity across the spherical representation.

**Inpainting for semantic editing.** Given a panoramic video, we identify and mask regions for modification. Next, we apply the denoising process to these regions, guided by user-provided text descriptions. Leveraging its understanding of spherical representations, the inpainted content naturally exhibits the properties of ERP projection.

**Outpainting for conventional videos.** Similar to the inpainting process, we first map the conventional video to the latent code and then mask the surrounding unseen panoramic regions. With user-provided text descriptions, we denoise the masked regions to generate corresponding content. Our pre-trained model maintains the spatial and temporal consistency for generated panoramic videos.

## 6 Conclusion

We present PanoWan, a text-based panoramic video generation framework that effectively lifts pre-trained diffusion model to the panorama. By integrating latitude-aware sampling, PanoWan addresses latitudinal distortions caused by equirectangular projection. Equipped with the rotated semantic denoising and the padded pixel-wise decoding, PanoWan achieves the seamless longitude transitions. To provide large-scale data for lifting representations from conventional videos to the panorama, we contribute the PANOVID dataset, offering high-quality and semantically rich 360° video data with annotated text descriptions. Extensive experiments demonstrate that PanoWan achieves state-of-the-art performance on text-based panoramic video generation and strong generalization across diverse zero-shot downstream tasks.

**Limitation.** While PanoWan benefits from the strong priors of its pretrained text-to-video models, it is also inherits a common challenge: the content forgetting problem often seen in such models. This issue is particularly evident when generating long videos due to limited temporal memory. We believe this challenge can be substantially alleviated through future advancements in memory-aware generation techniques (*e.g.*, video caching mechanisms).

**Acknowledgement.** This work is supported by National Natural Science Foundation of China under Grant No.62136001. Authors thank openbayes.com for providing computing resources.

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
