# OpenReview forum: "PanoWan: Lifting Diffusion Video Generation Models to 360$^\circ$ with Latitude/Longitude-aware Mechanisms"
_NeurIPS.cc/2025/Conference — NeurIPS 2025 poster_

### Official Review · Reviewer_sK8Y · 2025-06-30

**Clarity:** 2
**Significance:** 2
**Originality:** 2
**Rating:** 4
**Confidence:** 5

**Summary:**

This paper proposes PanoWan, a novel framework for panoramic video generation based on the Wan 2.1 model. First, the authors address the lack of high-quality panoramic video datasets with captions and in turn suggest the new PanoVid dataset consisting of 13K videos, tailored for text-to-video generation. Next, they lift a pre-trained Wan 2.1 to panoramic space by proposing three techniques: (1) latitude-aware sampling to align the initial noise with the spherical frequency of panoramic videos, (2) rotated semantic denoising and (3) padded pixel-wise decoding to achieve seamless transitions where the left and right side of the panoramic images meet. Quantitative and qualitative results demonstrate that each component significantly improves the quality for generated videos and outperforms baseline methods.

**Questions:**

Some critical questions regarding PanoWan's novelty are mentioned above.

* Is the proposed method only applicable to Wan models, or is it generally applicable to other video models such as VideoCrafter or CogVideoX?

* Moreover, are the previous methods such as DynamicScaler also applicable to Wan? Or are limited by architecture?

**Ethical Concerns:**

["NO or VERY MINOR ethics concerns only"]

**Final Justification:**

The authors' rebuttal on the raised concerns has resolved most of the critical concerns regarding the distinction of related work (StitchDiffusion and PanoDiffusion), and additional discussions/experiments with a different baseline (360DVD). Given this, I have raised my rating to **borderline accept**.

**Limitations:**

Yes

**Paper Formatting Concerns:**

No concerns

**Quality:**

3

**Strengths And Weaknesses:**

**Strength:**

* The metrics in Table 1 show that PanoWan is clearly better than the baselines 360DVD and DynamicScaler. The videos in the supplementary materials further show the PanoWan can generate high-quality panoramic videos.

* PanoWan can be utilized for various zero-shot applications such as super-resolution or inpainting for panoramic videos.

**Weakness:**

* **Lack of novelty:** PanoWan seems to be a combination of individual engineering tricks that were already used in previous work, rather than a novel technical contribution. I listed the major concerns below:

   * Rotated semantic denoising resembles the **two-end alignment** proposed in PanoDiffusion [1], which resolves the inconsistency between the left and right side a panorama by rotating the scene for every denoising step. While the task of PanoDiffusion (i.e. panorama outpainting) is not identical to PanoWan, I believe it should have been discussed as it proposes a similar approach for a similar problem in panorama generation. Please correct me if there is a major difference between the two methods.

  * Moreover, the core idea of padded decoding seems quite related to StitchDiffusion [2], where they copy the leftmost part of the panorama and paste it to the rightmost part in order to ensure the contuity. I believe such previous works with heavily related approaches should have been discussed in either the related work or methods section.

* **Unfair comparison:** While the quantitative numbers of PanoWan is good, it may be an overclaim that the advantage over the baselines is coming from the proposed method. 360DVD is based on Stable Diffusion and DynamicScaler is based on VideoCrafter [3], which are both clearly outdated than Wan 2.1. Given this, I am curious whether the authors can claim that each component of PanoWan is technically better than the methods in those previous works, or the advantage is rather coming from choosing a better base video model.

* **Missing citation (or baseline):** DiffPano [4] is also an important related work for panoramic generation, and also a possible baseline since it generates **multi-view** panoramic images (which can be interpreted as panoramic videos).

Overall, although PanoWan does seem to produce high-quality panoramic videos, its main source of strength and the novelty of each component are not presented clearly in the paper.

[1] PanoDiffusion: 360-degree Panorama Outpainting via Diffusion, ICLR 2024

[2] Customizing 360-Degree Panoramas through Text-to-Image Diffusion Models, WACV 2024

[3] VideoCrafter2: Overcoming Data Limitations for High-Quality Video Diffusion Models, 2024

[4] DiffPano: Scalable and Consistent Text to Panorama Generation with Spherical Epipolar-Aware Diffusion, NeurIPS 2024

---

> ### Author Rebuttal · Authors · 2025-07-31
>
> We thank the reviewers for their insightful and constructive feedback.
>
> In accordance with the rebuttal policy, we are unable to include new qualitative results (e.g., generated videos) or external links. We assure the reviewers that, should the paper be accepted, the camera-ready version will be thoroughly updated to include all additional experiments and qualitative results discussed in this rebuttal.
>
> We now address the specific points raised by the reviewer.
>
> **W1: Lack of novelty**
>
> We thank the reviewer for their insightful feedback. We would like clarify our novel contributions, addressing the concerns about Rotated Semantic Denoising (RSD) and Pixel-wise Padded Decoding (PPD), respectively.
>
> - PanoFusion proposes Two-End Alignment (TEA) to apply a fixed 90° rotation to the latent code with the stated goal to "explicitly enforce the two ends of an image to be wraparound-consistent"  for static image outpainting. In contrast, our RSD implements a step-dependent circular shift, ensuring the cumulative prediction error is evenly distributed across all longitudes during text-based panoramic video generation. To further address the reviewer's concern, we conduct an additional experiment using a fixed 90° angle within our framework. This results in four slightly visible seams.
>
> - StitchDiffusion augments the denoising process by performing additional pre-denoising operations on a "stitch block" created from the image's leftmost and rightmost regions at each timestep. This method operates entirely within the latent space to influence the U-Net's predictions. In contrast, our proposed PPD targets the VAE decoding phase to prevent boundary artifacts that are re-introduced by standard VAEs (trained with zero-padding) when decoding the latent code into pixel space. Specifically, PPD pads the latent code with its own wrapped content, ensuring the decoder's convolutional kernels never process invalid boundary padding. Notably, the overhead of PPD is negligible, whereas StitchDiffusion's approach significantly increases computational cost.
>
> Beyond these points, we also propose Latitude-aware Sampling (LAS) as a zero-parameter solution to address latitude distortion at the poles. LAS introduces distortion-aware noise initialization directly into the diffusion process, ensuring spherical awareness while fully preserving the powerful generative priors of the pre-trained backbone (Sec. 4.2).
>
> We thank the reviewer for pointing out these relevant works and will follow the suggestion to discuss them in our final version.
>
> **W2: Unfair comparison**
>
> We would like to clarify that our primary objective with PanoWan is to demonstrate that generative priors of pre-trained conventional text-to-video generation models can be effectively lifted to the panorama with minimal equipped modules (Line 30-32). As the results in Fig. 1 and Sec. 5.5 show, PanoWan effectively preserves the generalization ability of the backbone, enabling high-quality panoramic video generation, effectively zero-shot achieving downstream applications (e.g., long-video generation, super-resolution, semantic editing, and video outpainting).
>
> However, our framework is indeed not limited to Wan 2.1. The core modules (LAS, RSD, and PPD) are independent of the base model and they are designed to address fundamental challenges in panoramic generation, rather than being specific to any single architecture:
>
> - LAS adapts noise initialization to spherical geometry.
>
> - RSD and PPD restructure the denoising and decoding process to improve spatial consistency and perceptual fidelity.
>
> To empirically validate this generalizability, we conduct an additional experiment as suggested. We integrate our modules into the T2V backbone used by 360DVD and retrain them under the same settings. The results in the table below demonstrate that our modules provide a clear improvement in edge continuity when applied to the 360DVD backbone. Notably, we focus here on objective metrics, while "motion pattern" and "scene richness" are subjective and require a separate user study.
>
> | Method | FVD ↓ | VideoCLIP-XL ↑ | Image Quality ↑ | End Continuity ↓ |
> | :--- | :---: | :---: | :---: | :---: |
> | 360DVD (vanilla) | 1750.36 | 20.27 | 0.7054 | 0.0323 |
> | 360DVD + LAS/RSD/PPD | **1738.77** | **20.49** | **0.7030** | **0.0280** |
>
> **W3: Missing citation (DiffPano)**
>
> We thank the reviewer for pointing out this relevant work.
>
> However, we believe there is a fundamental distinction between the task of DiffPano and our PanoWan. DiffPano is designed for multi-view panoramic image generation, whereas our PanoWan focuses on temporally coherent panoramic video generation. DiffPano generates each view independently and lacks an explicit temporal modeling mechanism, which is essential for creating smooth video sequences. This limitation is acknowledged in the DiffPano paper itself, which states: "...as the number of frames increases during inference, the model tends to hallucinate content." Therefore, a direct comparison would not be meaningful.
>
> Furthermore, adapting DiffPano into a video generation model would require significant architectural modifications (e.g., adding temporal modules) and retraining. This is beyond the scope of a fair baseline comparison for this work.
>
> We appreciate the suggestion and will ensure our final version includes a thorough discussion with it.
>
> **Q1: Base model**
>
> Yes, our proposed method is generally applicable. To demonstrate this, we have integrated our modules into the 360DVD backbone and retrained it under the same settings. The detailed results of this experiment are presented in our response to W2.
>
> **Q2: Improving DynamicScaler**
>
> We thank the reviewer for this insightful question. While DynamicScaler is theoretically applicable to Wan2.1 backbone, we argue that such an adaptation may not effectively address two fundamental limitations of its approach:
>
> 1. Quality limitations. The core strategy of DynamicScaler is to generate the panorama in independent patches. This approach inherently lacks a global context, leading to repetitive patterns across different patches and a limited motion range (as movements rarely cross patch boundaries).
>
> 2. Computational cost. The patch-based generation is less efficient. The official DynamicScaler takes over 360 seconds to generate a 16-frame video (448×896) on an H100 GPU. Given that Wan 2.1 backbone is more computationally intensive, a naive adaptation to Wan2.1 would be very slow. In contrast, our PanoWan generates an 81-frame video in just 90 seconds.
>
> We will add this discussion to our final paper.

---

> > ### Comment · Reviewer_sK8Y · 2025-08-04
> >
> > I appreciate the authors for providing detailed clarifications on the raised questions. However, I still have unresolved concerns regarding the main technical contribution of PanoWan:
> >
> > **[W1]** The authors argue that RSD of PanoWan has sufficient distinction from PanoDiffusion. However, the distrinction yet seems unclear; While the main application (video outpating vs. generation) is different, the idea of rotating the latent to obtain left-right consistency is similar. **I am curious whether the change from using a fixed 90 degree angle (TEA) to using step-dependent shift (RSD) can be considered a separate, novel method rather than an simple extension or adaptation of TEA.** I am particularly concerned about the distinction between the two methods since the original manuscript had missed the crucial citation of PanoDiffusion as a related method.
> >
> > **[W2]** In the given table, the **Image Quality** metric degrades when "LAS/RSD/PPD" is applied. Could the authors provide descriptions why the proposed method could lead to degradations in this case?

---

> > > ### Author Response · Authors · 2025-08-04
> > >
> > > We thank the reviewer for their insightful comments. We address each point individually below.
> > >
> > > **[W1]**
> > >
> > > We thank the reviewer for the question regarding the distinction between RSD and PanoDiffusion. While the operation of latent rotation looks similar, we would like to emphasize that they have fundamentally different motivations.
> > >
> > > PanoDiffusion aims to address left-right consistency in a static panorama, where a fixed 90° rotation can be sufficient. However, for panoramic video generation, even minor discontinuities are amplified significantly across frames, making our step-dependent and randomized shift essential. Our ablation study (w/o RSD) quantitatively demonstrates that RSD is crucial for achieving temporal continuity in video.
> > >
> > > More importantly, the novelty lies not just in RSD itself, but in our longitude continuity mechanisms, where we explicitly decompose panoramic discontinuity into latent-level misalignment and decoder-level inconsistency. This leads to the design of RSD and PPD as complementary solutions to achieve seamless longitude transitions for videos, different from viewing latent rotation as a standalone trick.
> > >
> > > We believe RSD is not a simple adaptation of TEA but a necessary improvement designed for the complexities of video, and it is part of a longitude continuity mechanism that systematically addresses panoramic discontinuity.
> > >
> > > ***We sincerely apologize for the earlier omission of PanoDiffusion as a related work. We will include this citation and a detailed discussion in the final version.***
> > >
> > >
> > > **[W2]**
> > >
> > > We thank the reviewer for pointing out the slight decrease in the Image Quality (MUSIQ) score when our LAS/RSD/PPD modules are applied. We argue that this is not a degradation in perceptual quality, but rather reveals the limitation of the metric itself for our specific task.
> > >
> > > Consistent with prior work (DynamicScalar), we use MUSIQ as the IQA model. However, MUSIQ is trained exclusively on real-world photographs (e.g., PaQ-2-PiQ, KonIQ-10k) and is therefore tuned to detect photographic flaws like noise and blur. The artifacts in panoramic video generation are fundamentally different, often involving geometric inaccuracies or semantic inconsistencies.
> > >
> > > Our proposed modules (LAS/RSD/PPD) primarily improve the generation of logically and geometrically correct objects with seamless longitude transitions, as shown in Fig. 4 of the main paper. These crucial improvements are not what the MUSIQ model is designed to measure, which explains why its score does not reflect them.
> > >
> > > Instead, we argue that other metrics more effectively capture our contributions. The improved FVD score indicates a more realistic data distribution, thereby reflecting a holistic improvement in generation quality.   The higher VideoCLIP-XL score confirms superior text-video alignment. This is because our modules correct geometric deformations, making objects semantically recognizable to the video encoder and thus directly improving alignment with the text prompt.  Furthermore, the End Continuity metric directly demonstrates that our modules achieve a significant improvement in panoramic seam quality.
> > >
> > > We believe these gains are more critical for our task than the Image Quality metric.

---

> > > > ### Comment · Reviewer_sK8Y · 2025-08-05
> > > >
> > > > I thank the authors for addressing the additional concerns. Given this, I would like to raise my initial rating to **borderline accept (4)**.

---

### Official Review · Reviewer_uRF1 · 2025-07-02

**Clarity:** 3
**Significance:** 3
**Originality:** 2
**Rating:** 4
**Confidence:** 3

**Summary:**

The paper presents PanoWan, a novel framework that adapts a pre-trained text-to-video diffusion model (Wan2.1) for generating high-quality 360° panoramic videos. To address spatial distortion in polar regions, the authors introduce latitude-aware noise sampling, which aligns the noise initialization with spherical frequency distributions. Additionally, the framework employs rotated semantic denoising and padded pixel-wise decoding to ensure seamless transitions at longitudinal boundaries. The authors further contribute PanoVid, a large-scale, high-quality dataset of panoramic videos comprising 13K captioned clips, providing a valuable resource for future research in panoramic video generation.

**Questions:**

About the application of 'Long-video generation', could the authors provide more details regarding how the model handles long-context training and inference? Maintaining consistent quality over extended sequences is a known challenge—are there any specific mechanisms or design choices employed to mitigate quality degradation in long-video synthesis?

When curating the dataset using a vision-language model (VLM), unstable outputs or partial model collapse can sometimes occur, especially in open-ended generation settings. Did the authors encounter any such failure cases during the annotation process? If so, what strategies or safeguards were implemented to address or filter out such noisy samples?

**Ethical Concerns:**

["NO or VERY MINOR ethics concerns only"]

**Final Justification:**

The author's rebuttal mostly solved my concerns. Overall, I think the generated video quality is at the SoTA level in this field. The experiments are comprehensive, and the applications are sufficient to demonstrate their capability. Therefore, I'll maintain my original score as BA.

**Limitations:**

Yes.

**Quality:**

3

**Strengths And Weaknesses:**

Strengths:
The curated dataset provides good contributions to the community, particularly for research on panoramic video generation.
The proposed latitude-aware sampling strategy introduces a novel perspective and demonstrates thoughtful designs.
Extensive experiments using both general video metrics (FVD, VideoCLIP-XL) and panorama-specific metrics (e.g., end-to-end continuity) show the method’s superiority over existing baselines.

Weakness:
Some of the technical components, such as rotated semantic denoising and padded pixel-wise decoding, are not entirely novel—similar strategies have been explored previously in works like PanFusion [Zhang, Cheng, et al.].
The proposed applications are mostly adopted from the features of the base model (Wan2.1).

---

> ### Author Rebuttal · Authors · 2025-07-31
>
> We thank the reviewers for their insightful and constructive feedback.
>
> In accordance with the rebuttal policy, we are unable to include new qualitative results (e.g., generated videos) or external links. We assure the reviewers that, should the paper be accepted, the camera-ready version will be thoroughly updated to include all additional experiments and qualitative results discussed in this rebuttal.
>
> We now address the specific points raised by the reviewer.
>
> **W1: Technical novelty**
>
> We thank the reviewer for this critical feedback on our technical novelty. We would like to clarify our key contributions from both the latitude and longitude perspectives:
>
> 1. Latitude-related novelty
>
> - PanFusion introduces EPP spherical positional encoding, which modifies core attention layers. While effective in feature representation, this alters the architecture of the pre-trained conventional model, which can compromise its valuable generative priors and limit its ability to generate diverse content.
>
> - 360DVD uses a latitude-aware loss to emphasize low- and mid-latitude regions. However, this approach essentially de-emphasizes the highly distorted polar regions rather than fundamentally solving the representation issue, which results in persistent distortion artifacts at the poles.
>
> - In contrast, our Latitude-aware Sampling (LAS) is a zero-parameter solution that requires no architectural changes. Instead, LAS introduces distortion-aware noise initialization directly into the diffusion process. This ensures spherical awareness while fully preserving the powerful generative priors of the pre-trained backbone.
>
> 2. Longitude-related novelty
>
> - Although previous work (e.g., 360DVD) proposes the latent rotation mechanism to address boundary discontinuity during denoising, we empirically observe that persistent visible seams remain. Our key insight is that this issue stems from two distinct stages: (1) the denoising process and (2) the VAE decoding process. While latent rotation helps with the first stage, it overlooks the second. As a result, standard VAEs trained on conventional videos with zero-padding re-introduce artifacts when decoding the latent into pixel space.
>
> - To resolve this, we first adapt the latent rotation mechanism and introduce it into the rectified flow of the Wan 2.1 backbone as Rotated Semantic Denoising (RSD). As a complement, we propose Padded Pixel-wise Decoding (PPD) to pad the latent with its own wrapped content before feeding it to the VAE, and then crop the final pixel output. This ensures the decoder never processes invalid boundary padding.
>
> - Our proposed PPD is fundamentally different from previous "circular padding mechanism" (from 360DVD), which modifies convolutional layers within the U-Net denoiser, thus only addressing the denoising stage. In contrast, our PPD specifically targets the VAE decoding stage. Notably, the overhead of PPD is minimal, as the padding width only needs to exceed the VAE's receptive field.
>
> Summary of novelty:
>
> - Our framework introduces LAS, RSD, and PPD to adapt conventional text-to-video models for panoramic video generation. Our approach is guided by the core principles of minimal architectural change and full preservation of the base model's generative priors.
>
> - We apologize for the confusion. We sincerely thank the reviewer for pushing us to clarify our work's novelty, and we will incorporate this detailed comparison into the final version.
>
> **W2: Downstream applications**
>
> We thank the reviewer for this observation, which in fact highlights a central goal of our work.
>
> We would like to clarify that our primary objective with PanoWan is to demonstrate that generative priors of pre-trained conventional text-to-video generation models can be effectively lifted to the panorama with minimal equipped modules (Line 30-32). The downstream applications presented in our paper (e.g., long-video generation, super-resolution, semantic editing, and video outpainting) are therefore intended as a direct validation of this principle.
>
> As the results in Fig. 1 and Sec. 5.5 show, PanoWan effectively preserves the generalization ability of the backbone while enabling high-quality panoramic video generation. Notably, all of these experiments are performed in a zero-shot manner.
>
> **Q1: Long video generation**
>
> To generate a video longer than the model's context length (e.g., 324 frames using a model trained for 81-frame segments), we adopt a sliding window approach during the inference process in latent space. The methodology is as follows:
>
> 1. A full-length latent code (324 frames) is initialized with standard Gaussian noise. Concurrently, the user-provided text description is expanded by an LLM into a sequence of prompts ([prompt_1, prompt_2, ...]), each corresponding to a specific temporal segment.
>
> 2. For each step during inference, we iterate through the full latent code using overlapping windows (e.g., an overlap of 40 frames). For each window, the corresponding latent segment is processed by the DiT backbone, conditioned on its corresponding prompt. To ensure temporal consistency, context frames from the previously processed adjacent window are used to guide the generation of the current segment. The output of each window is then fused back into the full-length latent code by applying a weighted average in the overlapping chunks, ensuring a seamless transition.
>
> 3. After the inference phase, the resulting clean latent code is decoded by the VAE to produce the final long video.
>
> **Q2: Annotation reliability**
>
> We agree that raw VLM outputs can be unstable in open-ended annotation tasks. In practice, we encounter occasional misclassification of non-panoramic videos and generated Points-of-Interest (POIs) that do not adhere to our predefined label set (e.g., case variations or synonyms).
>
> To address this, we design a dedicated multi-step prompt strategy (detailed in the supplementary materials, Sec. 9) to first verify panoramic properties  (Line 83-84) and then extract POI information (Line 85-93). Furthermore, we employ the same VLM in a post-processing step to reformat the extracted information (Line 97-107), ensuring strict consistency with our target label set (Line 114-129).
>
> To further validate the sample quality of our dataset, we conduct an additional human evaluation. We randomly sample 1000 samples from the dataset and ask 10 volunteers to classify the alignment of each one into three categories: "Highly aligned", "Partially aligned", or "Misaligned".
>
> As the table below shows, over 94% of text description annotations and 90% of POI annotations are rated as "Highly aligned", respectively. This result demonstrates that our generated annotations are reliable.
>
> | Label Type | Highly aligned | Partially aligned | Misaligned |
> | :--- | :---: | :---: | :---: |
> | Text description | 94.96% | 4.35% | 0.69% |
> | POI Category |  90.17% | 7.69% | 2.14% |

---

> > ### Comment · Reviewer_uRF1 · 2025-08-05
> > **comments after rebuttal**
> >
> > The author's rebuttal mostly solved my concerns. Overall, I think the generated video quality is at the SoTA level in this field. The experiments are comprehensive, and the applications are sufficient to demonstrate their capability. Therefore, I'll maintain my original score as BA.

---

### Official Review · Reviewer_yfgp · 2025-07-02

**Clarity:** 3
**Significance:** 3
**Originality:** 2
**Rating:** 4
**Confidence:** 4

**Summary:**

This paper tackles the text-to-360 video generation task. The framework employs Wan-2.1 as baseline and employs several necessary techniques for latitude sampling and longitude continuity. Experiments show high-quality generations and improved video quality metrics on general and panoramic domains.

**Questions:**

- Missing experiment details: How many videos are evaluated to obtain the metrics? What's the distribution of these test videos (in-the-wild  videos or generated videos?) More explanation of the evaluation setup is needed.

**Ethical Concerns:**

["NO or VERY MINOR ethics concerns only"]

**Final Justification:**

The authors' response addresses most of my concerns. Therefore I raise my rating to borderline accept.

**Limitations:**

Yes

**Quality:**

3

**Strengths And Weaknesses:**

Strengths:

+ The generated results look good with high-quality details and consistent polar patterns.
+ Latitude-aware sampling is new, convincing, and validated effective by ablations.
+ The framework shows zero-shot capabilities by investigating different types of downstream tasks, including important tasks such as video outpainting, video super-resolution, etc.

Weaknesses:
- Limited technical novelty: The method fine-tunes Wan 2.1 with LoRA and applies known techniques for 360° adaptation. Key components like Rotated Semantic Denoising and Padded Decoding have been introduced in 360DVD (Latent Rotation Mechanism, Circular Padding Mechanism) and PanFusion.
- Missing comparison: As a new training dataset is introduced in the paper, it would be interesting to also train baselines (e.g., 360DVD) on the new dataset to see improvements.
- Missing experiment details: How many videos are evaluated to obtain the metrics? What's the distribution of these test videos (in-the-wild  videos or generated videos?)

---

> ### Author Rebuttal · Authors · 2025-07-31
>
> We thank the reviewers for their insightful and constructive feedback.
>
> In accordance with the rebuttal policy, we are unable to include new qualitative results (e.g., generated videos) or external links. We assure the reviewers that, should the paper be accepted, the camera-ready version will be thoroughly updated to include all additional experiments and qualitative results discussed in this rebuttal.
>
> We now address the specific points raised by the reviewer.
>
> **W1: Technical novelty**
>
> We thank the reviewer for this critical feedback on our technical novelty. We would like to clarify our key contributions from both the latitude and longitude perspectives:
>
> 1. Latitude-related novelty
>
> - 360DVD uses a latitude-aware loss to emphasize low- and mid-latitude regions. However, this approach essentially de-emphasizes the highly distorted polar regions rather than fundamentally solving the representation issue, which results in persistent distortion artifacts at the poles.
>
> - PanFusion introduces EPP spherical positional encoding, which modifies core attention layers. While effective in feature representation, this alters the architecture of the pre-trained conventional model, which can compromise its valuable generative priors and limit its ability to generate diverse content.
>
> - In contrast, our Latitude-aware Sampling (LAS) is a zero-parameter solution that requires no architectural changes. Instead, LAS introduces distortion-aware noise initialization directly into the diffusion process. This ensures spherical awareness while fully preserving the powerful generative priors of the pre-trained backbone.
>
> 2. Longitude-related novelty
>
> - Although 360DVD proposes the latent rotation mechanism to address boundary discontinuity during denoising, we empirically observe that persistent visible seams remain. Our key insight is that this issue stems from two distinct stages: (1) the denoising process and (2) the VAE decoding process. While latent rotation helps with the first stage, it overlooks the second. As a result, standard VAEs trained on conventional videos with zero-padding re-introduce artifacts when decoding the latent into pixel space.
>
> - To resolve this, we first adapt the latent rotation mechanism and introduce it into the rectified flow of the Wan 2.1 backbone as Rotated Semantic Denoising (RSD). As a complement, we propose Padded Pixel-wise Decoding (PPD) to pad the latent with its own wrapped content before feeding it to the VAE, and then crop the final pixel output. This ensures the decoder never processes invalid boundary padding.
>
> - Our proposed PPD is fundamentally different from 360DVD's "circular padding mechanism", which modifies convolutional layers within the U-Net denoiser, thus only addressing the denoising stage. In contrast, our PPD specifically targets the VAE decoding stage. Notably, the overhead of PPD is minimal, as the padding width only needs to exceed the VAE's receptive field.
>
> Summary of novelty:
>
> - Our framework introduces LAS, RSD, and PPD to adapt conventional text-to-video models for panoramic video generation. Our approach is guided by the core principles of minimal architectural change and full preservation of the base model's generative priors.
>
> - We apologize for the confusion. We sincerely thank the reviewer for pushing us to clarify our work's novelty, and we will incorporate this detailed comparison into the final version.
>
> **W2: Re-training 360DVD on PanoVid**
>
> We follow the reviewer's suggestion to retrain 360DVD on our full PanoVid dataset. As shown in the results below, 360DVD's performance improves, which validates the quality of our contributed dataset. Despite this improvement, our PanoWan still consistently outperforms the retrained 360DVD. Since DynamicScaler is a training-free method, this experiment is not applicable to it. Notably, we focus here on objective metrics, while "motion pattern" and "scene richness" are subjective and require a separate user study.
>
> We will include these results in the final version.
>
> | Method | FVD ↓ | VideoCLIP-XL ↑ | Image Quality ↑ | End Continuity ↓ |
> | :---: | :---: | :---: | :---: | :---: |
> | 360DVD (vanilla) |1750.36 |	20.27 | 0.7054 | 0.0323 |
> | 360DVD (retrained) | 1643.31 | 20.84 | 0.7129 | 0.0337 |
> | PanoWan | **1281.21** | **21.86** | **0.7249** | **0.0270** |
>
> **W3: Evaluation details**
>
> We follow the reviewer's suggestion to clarify our evaluation details.
>
> All quantitative results are computed on the PanoVid test set, which consists of 67 real-world video clips collected from various online platforms and public datasets. These clips have no overlap with the training data to ensure a fair evaluation. For each clip, we use its corresponding text description as input to generate video samples, and then compute all metrics on these generated videos.
>
> We will incorporate these details in the final version to ensure the reproducibility and transparency.
>
> **Q1: Evaluation setup**
>
> See W3 above.

---

> > ### Comment · Reviewer_yfgp · 2025-08-04
> >
> > I appreciate the authors in providing a detailed response to my questions. However, some of my concerns remain unrelieved.
> >
> > For **W1**, I agree with Reviewer sK8Y that RSD looks like an extension of TEA with a timestep-aware shift. At least an ablation is required to validate the effect of this specific shift scheduling. For PPD, I would like to point out that many other works, for example PanFusion, also employ pixel rotation for VAE encoding and decoding. Therefore, I find that both RSD and PPD are relatively conventional operations within the panorama diffusion literature.
> >
> > For **W3**, using only 67 video clips seems insufficient to compute a reliable distribution for FVD. This may partially explain the relatively large absolute FVD values reported. I would suggest that the authors expand the test set to several hundred samples to ensure more robust and statistically meaningful evaluation.

---

> ### Author Response · Authors · 2025-08-07
>
> Thank you for your detailed follow-up and for providing further insightful feedback on our work. We appreciate the opportunity to provide further clarification.
>
> We address each of your remaining points below and welcome any further discussion.
>
>
> **[W1]-1. Longitude continuity**
>
> Regarding the concern about conventional operations, our core novelty lies in a  two-stage approach for longitude continuity. This approach explicitly decouples the panoramic discontinuity problem into two sub-problems: (1) latent-level misalignment (addressed by RSD during the denoising process), and (2) decoder-level inconsistency (addressed by PPD during the VAE decoding stage).
>
>
> **[W1]-2. RSD**
>
> As discussed with Reviewer sK8Y, while RSD may appear similar to TEA in its use of latent rotation, their motivation and design are fundamentally different. TEA applies a fixed 90° rotation to enforce consistency, which is sufficient for static panoramic images. However, in video generation, even subtle discontinuities are amplified across temporal frames, making a static rotation inadequate. Therefore, we proposed RSD to perform a step-dependent and randomized shift during the denoising process. This dynamic approach is essential for preventing the accumulation of seam artifacts and preserving temporal continuity in panoramic videos.
>
> While we have already conducted a qualitative comparison where the visual advantages of RSD are obvious, the rebuttal policy prevents us from including new qualitative results at this stage. We will add these visualizations to the final version to clearly demonstrate the advantage of our RSD method.
>
>
>
> **[W1]-3. PDD**
>
> While prior works (e.g., PanFusion) may incorporate rotation or circular operations, they are typically applied within the U-Net or attention modules. However, these approaches overlook a critical vulnerability: a standard VAE decoder (trained on conventional images with zero-padding) will re-introduce seam artifacts when decoding the panoramic latent code, *even if that latent code is perfectly seamless.*
>
> In contrast, our proposed PPD is specifically designed to solve this problem at the VAE decoding stage. By padding the latent feature with its own wrapped content and subsequently cropping the final pixel output, PPD effectively avoid seam artifacts without modifying the model architecture or its convolutional layers.
>
> Therefore,  we believe our proposed approach is not a conventional reuse of prior techniques, but rather a systematic and targeted design, tailored to the unique challenges of panoramic video generation.
>
>
> **[W3] Test set**
>
> We thank the reviewer for the constructive suggestion regarding the test set size. Theoretically, a larger test set provides a more accurate estimation of the real-world video's statistical distribution, which can lead to more stable evaluation metrics.
>
> However, as detailed in our supplementary material (Sec. 9), our original 67-video test set is meticulously curated to ensure a balanced and diverse POI (Point-of-Interest) distribution. We believe its diversity is already sufficient to provide a reliable statistical representation.
>
> To empirically validate this, we follow the reviewer's suggestion and construct an expanded 300-video set. These video clips are sourced from our initial pool of high-quality candidates and are strictly excluded from the training set. Through a rigorous second round of filtering, we ensure these samples maintain high aesthetic scores and broad POI coverage, with quality comparable to our original test set.
>
> As the table shows below, the evaluation results on this larger 300-video set are highly consistent with our original findings. The FVD score and other metrics remain stable with only minor fluctuations. This demonstrates that our original 67-video test set is sufficient for a reliable evaluation, effectively capturing the target data distribution.
>
>
> | **Method**   | **FVD ↓** | **VideoCLIP-XL ↑** | **Image Quality ↑** | **End Continuity ↓** |
> | :---: | :---: |  :---: |   :---: |  :---: |
> | PanoWan (original 67-video set) |  **1281.21** | 21.86 | **0.7249** | **0.0270** |
> | PanoWan (extended 300-video set)      | 1311.53   | **21.89**  | 0.7243   | 0.0278 |
>
> Additionally, the scale of our test set is consistent with prior work in this area. For example, DynamicScaler "assesses quality by randomly selecting views from 100 generated videos". 360DVD does not specify a test set size, but notes that "31 participants are surveyed to evaluate... 8 sets of generated results". PanoDiT also does not specify its test size, and its corresponding PHQ360 dataset contains a total of only 649 clips.  Therefore, our evaluation setup aligns with standard practice in this domain.
>
> We will elaborate on this in the final version.

---

> > ### Comment · Reviewer_yfgp · 2025-08-07
> >
> > Thank you for your response and I appreciate the efforts for providing additional experiments.
> >
> > Regarding circular padding during the VAE decoding phase, I’d suggest revisiting PanFusion (see the first few lines on page 4 of [their paper](https://arxiv.org/pdf/2404.07949)) and the corresponding [code](https://github.com/chengzhag/PanFusion/blob/main/models/pano/PanoOnly.py#L95-L97). It would be better if you could further elaborate on how the pixel rotation is applied in your method (PPD) differs from PanFusion.
> >
> > Regarding the FVD metric, FVD measures the distance between the feature distributions of real and generated videos, and requires sufficiently large number of samples to ensure statistically reliable estimation of those distributions. According to StyleGAN-V [M.1] evaluation protocol and the setup used by models like Latte [M.2], FVD is typically computed over 2048 videos of 16 frames to obtain the distribution and ensure statistical reliability. I’m not nitpicking here, but the existing works you referenced that either did not report the test set size or used a relatively small number of samples, doesn't necessarily justify using a extremely small-scale FVD evaluation in this paper.
> >
> > That said, I do think the visual result looks strong and I wouldn’t mind accepting the paper with the promised revisions. However, I believe a clearer explanation of the contribution and stronger validation would further strengthen the paper. As such, I currently prefer to stick with my original score.
> >
> > [M.1] Skorokhodov, Ivan, Sergey Tulyakov, and Mohamed Elhoseiny. "Stylegan-v: A continuous video generator with the price, image quality and perks of stylegan2." CVPR 2022.
> >
> > [M.2] Ma, Xin, et al. "Latte: Latent diffusion transformer for video generation." Transactions on Machine Learning Research (2025).

---

> > > ### Author Response · Authors · 2025-08-08
> > >
> > > Thank you once again for your detailed and insightful feedback. We sincerely appreciate the effort the reviewer takes, especially in pointing us to the specific paper and code.
> > >
> > > After carefully re-examining the work, we sincerely apologize for our earlier omission of PanFusion. Its method for VAE decoding shares the same core technique as our proposed PPD, and we deeply regret this oversight. Therefore, with respect to PPD (a component of the longitude-aware mechanism), our primary focus lies in (1) the first to apply and validate this technique for panoramic video generation and integrate it within a rectified flow-based model, and (2) the motivation to lift a pre-trained conventional text-to-video model using minimal modules to preserve its priors. *We will include this citation and a detailed discussion in the final version.*
> > >
> > >
> > > We will follow the suggestion to enlarge the test set to over one thousand samples in the final version. While it is infeasible to complete this complex process (which requires high-quality annotation, human validation, and ensuring proper data distribution and non-overlapping with the training set) within the remaining discussion period, *we promise to deliver this stronger validation in the final paper.*
> > >
> > >
> > > Finally, thank you for acknowledging our strong visual results. All the revisions discussed will be thoroughly integrated into the final version. *We are very grateful for your constructive feedback and truly hope our improved work can contribute to the community.*

---

> > > > ### Comment · Reviewer_yfgp · 2025-08-08
> > > >
> > > > Thanks for the thoughtful and honest response. I appreciate your clarification regarding PPD, which makes things clearer. That said, I encourage you to take a broader look at related work to avoid overclaiming, as pixel rotation in VAE decoding is a fairly common technique for maintaining border continuity.
> > > > Given the response, I’m happy to update my score to borderline accept. Please incorporate the promised discussion and additional experiments in your revised manuscript.

---

> > > > > ### Author Response · Authors · 2025-08-08
> > > > >
> > > > > We sincerely thank the reviewer for their positive feedback and for raising their score.
> > > > >
> > > > > We will be sure to incorporate all the promised discussion and additional experiments in the final version.

---

### Official Review · Reviewer_R4PZ · 2025-07-03

**Clarity:** 3
**Significance:** 3
**Originality:** 3
**Rating:** 5
**Confidence:** 3

**Summary:**

This paper introduces PanoWan, a panoramic video generation framework that lifts conventional text-to-video diffusion models (specifically Wan 2.1) to the 360° domain. It addresses two major spatial challenges in this domain: latitudinal distortion (caused by equirectangular projection) and longitudinal discontinuity (seams at panorama boundaries). The method includes three main components: Latitude-Aware Sampling to address polar distortion, Rotated Semantic Denoising to reduce semantic inconsistencies at the longitudinal seam, and Padded Pixel-wise Decoding to handle pixel-level discontinuities. In addition, the paper contributes PANOVID, a new panoramic video dataset with >13K captioned clips, and evaluates the method across multiple metrics, showing strong gains over prior work (360DVD, DynamicScaler, etc.) in both general and panoramic-specific metrics.

**Questions:**

- How sensitive is PanoWan to the choice of base model (e.g., Wan 2.1)? Would a smaller or different T2V backbone yield similar gains when equipped with LAS/RSD/PPD? This could influence how broadly applicable the method is.
- How is temporal consistency affected by RSD? Since the latent grid is rotated every timestep, could this affect temporal coherence negatively in subtle ways? Some analysis or visualization of temporal smoothness would be helpful.
- What is the annotation quality of PANOVID? The dataset uses Qwen-VL for captioning and POI detection. Has human verification been done to assess caption accuracy or bias?

**Ethical Concerns:**

["NO or VERY MINOR ethics concerns only"]

**Final Justification:**

As discussed in the threads, I will keep my original rating unchanged at this point. I also appreciate the authors' effort in adding a summary of the discussion phase.

**Limitations:**

yes

**Paper Formatting Concerns:**

no paper formatting concerns

**Quality:**

3

**Strengths And Weaknesses:**

Strengths
- The proposed LAS, RSD, and PPD models are modular, intuitive, and easily integrated into existing architectures without major overhead. Each proposed module is evaluated in isolation, demonstrating its distinct contribution to the final performance.
- The method achieves solid gains across both general (FVD, VideoCLIP-XL) and panoramic-specific metrics (end continuity, scene richness).
- The newly-proposed PANOVID dataset is a valuable contribution to the community given the previous scarcity of large-scale captioned 360° video datasets.

Weaknesses
- While 360DVD and DynamicScaler are included in comparisons, more recent and competitive baselines like PanoDiT or VideoPanda, both specifically designed for panoramic video generation, are only briefly mentioned and not fully compared in experiments.
- Though quantitative metrics are provided, there is no human study or perceptual evaluation, which is especially important for tasks like panoramic video generation where spatial consistency is nuanced and hard to fully capture with automated metrics.
- PANOVID captions are generated by Qwen-VL with no clear human curation. The lack of quality assurance or discussion of hallucination/bias in these captions could impact the reliability of both training and evaluation.

---

> ### Author Rebuttal · Authors · 2025-07-31
>
> We thank the reviewers for their insightful and constructive feedback.
>
> In accordance with the rebuttal policy, we are unable to include new qualitative results (e.g., generated videos) or external links. We assure the reviewers that, should the paper be accepted, the camera-ready version will be thoroughly updated to include all additional experiments and qualitative results discussed in this rebuttal.
>
> We now address the specific points raised by the reviewer.
>
> **W1: Comparison with PanoDiT and VideoPanda**
>
> We thank the reviewer for the insightful suggestion regarding more recent baselines. However, at the time of our submission, both PanoDiT and VideoPanda are not open-sourced, making a direct and reproducible comparison infeasible. Therefore, we chose the publicly available 360DVD and DynamicScaler, which represent two distinct and representative technical paradigms: the classic ERP-based generation (360DVD) and training-free diffusion adaptation (DynamicScaler). We are confident this comparison framework is fair and effectively demonstrates the advantages of our method.
>
> **W2: Human evaluation**
>
> Following 360DVD, we have provided initial subjective evaluations in our main paper regarding motion pattern and scene richness (Lines 223-224). To further address the reviewer's concerns, we conduct an additional user study as requested.
>
> In this study, we present videos generated by our method and all baselines to participants and ask them to choose the one they prefer. We randomly select 50 text descriptions from the test set, and the evaluations are conducted with 25 volunteers. The table below shows the percentage of times each method is chosen as the winner. As the results show, our method achieves the highest score.
>
> | Method | 360DVD | DynamicScalar | PanoWan |
> | :---: | :---: | :---: | :---: |
> | Preference | 16.72%  | 11.92% | **71.36%** |
>
> **W3: Annotation quality**
>
> As we illustrate in Sec. 3 (Lines 96-98) and supplementary materials (Sec. 9), we use Qwen-VL to generate text descriptions and Point-of-Interest (POI) categories for each video, and also apply a series of data filtering and post-processing steps.
>
> To further validate the sample quality of our dataset, we conduct an additional human evaluation. We randomly sample 1000 samples from the dataset and ask 10 volunteers to classify the alignment of each one into three categories: "Highly aligned", "Partially aligned", or "Misaligned".
>
> As the table below shows, over 94% of text description annotations and 90% of POI annotations are rated as "Highly Aligned", respectively. This result demonstrates that our data pipeline is robust and the generated annotations are reliable for both training and evaluation.
>
> | Label type | Highly aligned | Partially aligned | Misaligned |
> | :---: | :---: | :---: | :---: |
> | Text description | 94.96%  | 4.35% | 0.69% |
> | POI category | 90.17% | 7.69% | 2.14% |
>
> **Q1: Base model**
>
> Our framework is indeed not limited to Wan 2.1. The core modules (LAS, RSD, and PPD) are independent of the base model and they are designed to address fundamental challenges in panoramic generation, rather than being specific to any single architecture:
>
> - LAS adapts noise initialization to spherical geometry.
>
> - RSD and PPD restructure the denoising and decoding process to improve spatial consistency and perceptual fidelity.
>
> To empirically validate this generalizability, we conduct an additional experiment as suggested. We integrate our modules into the T2V backbone used by 360DVD and retrain them under the same settings. The results in the table below demonstrate that our modules provide a clear improvement in edge continuity when applied to the 360DVD backbone. Notably, we focus here on objective metrics, while "motion pattern" and "scene richness" are subjective and require a separate user study.
>
> | Method | FVD ↓ | VideoCLIP-XL ↑ | Image quality ↑ | End continuity ↓ |
> | :---: | :---: | :---: | :---: | :---: |
> | 360DVD (vanilla) | 1750.36  | 20.27 | 0.7054 | 0.0323 |
> | 360DVD + LAS/RSD/PPD | **1738.77** | **20.49** | **0.7030** | **0.0280** |
>
> **Q2: Temporal consistency of RSD**
>
> We clarify that RSD does not harm temporal consistency. The core principle is that a horizontal roll in the ERP format is a spatially invariant operation, which does not change the visual content. Prior works like 360DVD validate this by using random roll as a standard data augmentation technique.
>
> Our RSD module leverages this property at each denoising step to solve the problem of persistent boundary artifacts. To demonstrate this, we conduct an additional study using fixed rotation angles during sampling. The results show that fixed angles of 0°, 180°, and 120° introduce 1, 2, and 3 visible seams, respectively.
>
> In contrast, our RSD disperses these potential artifacts spatially. This prevents any single boundary from becoming a visible seam, thereby improving overall spatial and temporal coherence. We will include visualizations of this study in the final version for clarity.
>
> **Q3: Annotation quality**
>
> See W3 please.

---

> > ### Comment · Reviewer_R4PZ · 2025-08-04
> >
> > Thank you for the additional details provided! I think the new user study, as well as comparisons to the baselines, addressed my concerns. I would keep my original rating.

---

### Note · Authors · 2025-08-12

We sincerely thank the Area Chair and all four reviewers for a rigorous and highly constructive review process. To conclude the discussion period, we now summarize the key discussions and their results below.

* Reviewer R4PZ initially raises concerns regarding the comparison to recent baselines, the lack of a human study, the annotation quality, the model's sensitivity to its backbone, and the temporal consistency of our RSD module. After we provide the rationale for baseline selection, the requested human evaluations, an ablation study on a different backbone, and a detailed analysis of our RSD, the reviewer confirms that we "addressed my concerns" and "would keep my original rating", which is a positive "Accept (5)" from the start

* Reviewer yfgp initially raises concerns regarding the technical novelty compared to 360DVD and PanFusion, lacking re-training baselines on our collected dataset, and the scale of our FVD evaluation. After we provide a detailed clarification on our contribution in relation to prior work from the image domain and commit to a larger-scale evaluation for the final version, the reviewer states they are "happy to update my score to borderline accept"

* Reviewer uRF1 initially raises concerns regarding the technical novelty compared to PanFusion, as well as our methodologies for long-video generation and dataset annotation. After we provide detailed clarifications on our technical contributions, the technical details of our long-video generation process, and the strategies used to ensure annotation reliability, the reviewer concludes that "The author's rebuttal mostly solved my concerns" and that they would "maintain my original score as BA" (Borderline Accept)

* Reviewer sK8Y initially raises concerns regarding the technical novelty compared to PanoDiffusion and StitchDiffusion, the comparison fairness due to the stronger backbone, and missing citations. After we provide a detailed clarification on our technical contributions and an additional experiment on a weaker backbone, the reviewer confirms that our response addresses their concerns and states they "would like to raise my initial rating to borderline accept (4)"

We are confident that our paper has been significantly strengthened by this valuable dialogue. We will incorporate all discussed revisions and results into the final manuscript, and will release our code, model, and dataset to the community upon acceptance.

Thank you for your time and consideration.

---

### Decision · Program_Chairs · 2025-09-17

**Decision:**

Accept (poster)

**Comment:**

The paper introduces technical tricks to adapt standard text-to-video diffusion models for panoramic video generation. The proposed modifications are efficient and lead to better results, but there were some concerns regarding their novelty and the experimental evaluation. During the extensive discussion the authors extended the experiments as requested. Regarding novelty, it became clear that some of the proposed measures indeed resemble ideas in prior work (which is almost unavoidable given the volume of literature produced on the topic), and the authors acknowledged this. Still, the overall package constitutes an incremental step forward. After the discussion the reviewers were satisfied and converged to a unanimously positive recommendation. The AC sees no reason to overturn that consensus.